# Genetic and Functional Characterization of Congenital HCMV Clinical Strains in Ex Vivo First Trimester Placental Model

**DOI:** 10.3390/pathogens12080985

**Published:** 2023-07-27

**Authors:** Deborah Andouard, Valentin Tilloy, Elodie Ribot, Melissa Mayeras, Daniel Diaz-Gonzalez, Chahrazed El Hamel, Fabienne Piras-Douce, Nathalie Mantel, Sophie Alain

**Affiliations:** 1Bacteriology-Virology-Hygien Department, National Reference Center for Herpesviruses, Centre Hospitalier Universitaire de Limoges, 87000 Limoges, France; valentin.tilloy@unilim.fr (V.T.); elodie.ribot@chu-limoges.fr (E.R.); melissa.gomes@etu.unilim.fr (M.M.); 2RESINFIT—Antimicrobials: Molecular Supports of Resistances and Therapeutic Innovations, UMR Inserm 1092, University of Limoges, 87000 Limoges, France; daniel.diaz-gonzalez@chu-limoges.fr; 3Sanofi Vaccines R&D, 69280 Marcy-l’étoile, France; fabienne.piras-douce@sanofi.com (F.P.-D.); nathalie.mantel@sanofi.com (N.M.); 4UF9481 Bioinformatics Department, CHU Dupuytren, 87000 Limoges, France; 5UF8843 Medical Genomics Department, CHU Dupuytren, 87000 Limoges, France; 6Pediatric Department, Mother and Child Biobank (CB-HME), Hôpital de la Mère et de l’enfant, CHU Limoges, 87000 Limoges, France; chahrazed.elhamel-belili@chu-limoges.fr

**Keywords:** congenital cytomegalovirus, clinical strains, placenta, viral kinetics, mutations

## Abstract

Human cytomegalovirus (HCMV) is the leading cause of congenital viral infection, leading to a variety of symptoms in the unborn child that range from asymptomatic to death in utero. Our objective was to better understand the mechanisms of placental infection by HCMV clinical strains, particularly during the first trimester of pregnancy. We thus characterized and compared the replication kinetics of various HCMV clinical strains and laboratory strains by measuring viral loads in an ex vivo model of first trimester villi and decidua, and used NGS and PCA analysis to analyze the genes involved in cell tropism and virulence factors. We observed that first trimester villi and decidua are similarly permissive to laboratory and symptomatic strains, and that asymptomatic strains poorly replicate in decidua tissue. PCA analysis allowed us to segregate our clinical strains based on their clinical characteristics, suggesting a link between gene mutations and symptoms. All these results bring forth elements that can help better understand the mechanisms that induce the appearance of symptoms or in the congenitally infected newborn.

## 1. Introduction

Human cytomegalovirus (HCMV) is a worldwide member of the ß-herpesvirus family that can cross the placenta during pregnancy. Approximately 11% of live-born infants with congenital CMV (cCMV) infection are symptomatic [1]. Congenital CMV can cause mortality or a variety of symptoms, such as hearing loss or microcephaly [2]. Virus transmission occurs throughout pregnancy, but disease is more severe when primary maternal infection occurs during the first trimester of gestation. HCMV is capable of replicating in a variety of placental cells, including trophoblast progenitors in the chorion [3], infecting cell column cytotrophoblasts, and impairing the invasiveness of cells that form the anchoring villi. To date, the exact mechanisms of HCMV infection remain poorly understood.

Due to the strict host specificity of HCMV and the hemochorial structure of the human placenta, it is not possible to use animal models to study cCMV. Current in vitro models, consisting of cell monolayers, have some limitations and do not represent the complexity of the placenta. Several ex vivo models, using placental biopsies, have been developed to study the impact of HCMV in the tissue. These models allow the study of events in different placental compartments, such as the anchoring villi [4], the floating villi [5,6], and the decidua [7]. In these ex vivo models, mostly laboratory strains or BAC derivative strains (AD169, TB40/E, Merlin, VR1814) have been used, with the limitation of having been passaged many times or having lesions in the genome that could affect cell tropism. Very few low-passaged clinical strains are available for tissue infection and no data are available in cCMV ex vivo models; only biopsies of in utero infected placenta have been studied [8].

Currently, there is a lack of knowledge on the relationships between the timing of HCMV infection, clinical strain characteristics, and impact on the maternal and fetal side of the placenta. To better understand the patterns of HCMV infection in the first trimester placenta, we aimed to characterize different cCMV strains, which were obtained from the French National Reference Center (NRC) for Herpesviruses collection, by Next Generation Sequencing (NGS) and placental infection on the maternal and fetal side of the placenta. We selected nine low-passaged infectious clinical strains and produced cell-free stock virions to infect an ex vivo model of first trimester villi and decidua. Among these clinical strains, four symptomatic and two asymptomatic strains were able to infect both tissues, showing different behavior in terms of kinetics in the tissues.

## 2. Materials and Methods

### 2.1. Cell Lines and Viruses

#### 2.1.1. Cell Culture

Human embryonic lung fibroblast (HEF) (MRC-5 cells, BioMérieux, Craponne, France) cells and an epithelial cell line (ARPE-19, ATCC^®^ CRL-2302, Molsheim, France) were cultured in minimum essential medium containing L-glutamine (MEM Glutabio, Eurobio, Courtaboeuf, France) and minimal essential medium (DMEM F12, Fisher Scientific, Illkirch, France), respectively, and were both supplemented with 10% fetal bovine serum (Eurobio scientific, Courtaboeuf, France), 50 μg/mL penicillin, and 10 μg/mL gentamicin.

#### 2.1.2. Viruses

##### Endotheliotropic Strain TB40/E and Production of Clinical Strains

TB40/E was provided by Stephane Chavanas (UMR 1043, CPTP, Toulouse, France). HCMV clinical strains were obtained from NRC for Herpesviruses collection (CRBiolim, Limoges University Hospital).

Primary biological samples were thawed, filtered, and seeded on ARPE-19 monolayer in a small culture flask. All clarified virus stocks strains were obtained after passages on confluent ARPE-19 monolayers in 25 cm^2^, then on 75 cm^2^, and twice on 175 cm^2^ culture flasks, with up to 90–100% cytopathic effect. The culture supernatant containing the free virus was collected and pooled with cell-associated virions obtained after cell lysis by a −80 °C freeze/thaw cycle, then clarified by centrifugation at 3500 rpm for 10 min and stored at −80 °C. Infectious viral titers were determined on MRC-5 as foci forming units per milliliter (pfu/mL).

##### Production BADrUL131-Y4

BADrUL131-Y4 [9] was derived from a BAC clone (provided by Thomas Shenk, Molecular Biology Department, Princeton University, Princeton, NJ, USA) of the AD169 HCMV strain, in which the *UL131* ORF was repaired and the *UL21.5* region of the virus was replaced with a marker cassette containing the GFP coding region. BAC were electroporated into MRC-5 fibroblasts, and the resulting virus was amplified in ARPE-19 cells. Supernatant and cell-associated virus from cell lysate (freeze/thaw cycles) were pooled and clarified by centrifugation. Virus titers were determined by plaque assay in both ARPE-19 and MRC-5 cells.

### 2.2. Next Generation Sequencing and Data Analysis

Viral DNA was isolated using Hirt method from low-passage (<4) isolates [10], depleted of human DNA using the NEBNextR Microbiome DNA Enrichment kit (New England Biolabs, Ipswich, MA, USA), purified by magnetic beads (Agencourt AMPure XP), and then fragmented using the Ion Xpress Plus DNA Fragment Library Preparation kit (ThermoFisher Scientific, Waltham, MA, USA). Barcode adapters were ligated to the end of the fragments and 250 bp fragments were collected. These libraries were PCR amplified and then sequenced using IonTorrent technology on either the Proton or S5 sequencer with the Ion Sequencing kit (ThermoFisher Scientific, Waltham, MA, USA).

The bioinformatic analysis was performed using the ASPICmv workflow (https://gitlab.com/vtilloy/aspicmv, accessed on 20 March 2023). This pipeline follows the same methodology as described in ASPICov workflow v1.1.5c [11], without genotyping steps. The reference used was TB40/E (KF297339.1), and genes of interest were extracted using bedtools v2.27.1 (Quinlan laboratory, University of Utah, Salt Lake City, UT, USA) and custom awk commands. The resulting nucleotide consensus sequences are available under GenBank accession number 2691551.

The intersections of variation (SNPs and indels) between all samples were checked using bedtools v2.27.1. These variations, first in the whole genome and then in genes of interest, were extracted, sorted, counted, and collected in a table using custom awk commands. Counts were normalized according to gene size, then PCA and a parallel coordinate chart were performed using a custom Python script (https://gitlab.com/vtilloy/utils_python/PCA.py, accessed on 6 April 2023). Consensus sequences were processed (reverse complement where necessary, hypervariable region extraction and amino acid conversion) using bedtools v2.27.1 and custom scripts (https://gitlab.com/vtilloy/utils_bash/reverse_complement.sh and https://gitlab.com/vtilloy/utils_bash/dna2protein.sh, accessed on 24 March 2023). These sequences were then aligned to each other and to extracted references (from Genbank) using MUSCLE v3.8.1551. Phylogenetic trees were constructed from the alignments using Iqtree v1.6.12(Center for Integrative Bioinformatics Vienna, University of Vienna, Vienna, Austria) (1000 bootstraps) and visualized using Archaeopteryx v0.9914 (Burnham Institute for Medical Research, La Jolla, CA, USA).

These analyses were performed for all glycoprotein genes (*UL55*, *UL73*, *UL74*, *UL75*, *UL100*, and *UL115*), pentameric subunits (*UL128-130-131A)*, and virulence factors (*UL11*, *UL20*, *UL35*, *UL37*, *UL48A*, *UL57*, *UL88*, *UL104*, *UL144*, *US9*, *US14*, *US19*, *US22*, and *US28*).

### 2.3. Ex Vivo Model of cCMV

#### 2.3.1. Placenta Collection

Placentas were collected from voluntary abortions in the first trimester (6–14 weeks of gestation), with the women’s consent, in collaboration with the Biological Resources Center of the Mother and Child Hospital of Limoges, France (Mother and Child Biobank CB-HME/CRBioLim, Limoges, France, certified ISO: 20387). Prior to placental sampling, HCMV IgG serostatus was determined by chemiluminescent enzyme assay using the Liaison^®^ XL analyzer (Diasorin). Floating villi and decidua biopsies (5 mm^3^) were washed with 0.9% saline solution and cultured in minimum essential medium (MEM, Eurobio) supplemented with 10% fetal bovine serum (Gibco, Life Technology), 50 μg/mL penicillin, and 10 μg/mL gentamicin.

#### 2.3.2. Placenta Model

The ex vivo model was used as previously described [12]. Briefly, HEF cells were infected with cell-free viral stocks of BAD-CMV, TB40/E, or clinical strains at a multiplicity of infection (MOI) of 1. After seven days of incubation at 37 °C (5% CO_2_), a sponge (Spongostan dental™, NewPharma, Belgium) with a placental villi or decidua explant was added to each well and incubated at 37 °C (5% CO_2_). After a further seven days, the sponges and explants were transferred to new empty plates with fresh medium.

#### 2.3.3. Viral Kinetics in Tissue

To determine viral infection and replication kinetics, samples from each condition were collected and measured at different times after infection (day 0, 4, 7, 10, and 13 after plate change). Explants were lysed in Proteinase K (Qiagen, Germany) solution containing 200 nM Tris and 10% sodium dodecyl sulfate (SDS) at 56 °C until complete digestion. DNA extraction was then performed using NucliSENS^®^ technology on an EasyMag instrument (BioMérieux, France) according to the “specific B” protocol. Duplex quantitative PCR for HCMV UL83 and for albumin genes was performed as described [12].

### 2.4. Statistical Analysis

Differences between kinetics in villi and decidua and between early and late first trimester explants were analyzed using GraphPad Prism software (GraphPad Software, San Diego, CA, USA). A comparative analysis was performed using a two-way ANOVA test. Statistically significant differences were defined by a *p*-value less than 0.05.

## 3. Results

### 3.1. Clinical Strains

In the NRC database, we found 36 patients with cCMV (with or without symptoms); clinical data were missing for 17 of them, and we were unable to obtain viral isolates for 9 subjects. Therefore, we obtained 6 viral isolates from patients with symptoms at birth, called “symptomatic,” and 3 isolates from neonates without symptoms at birth, called “asymptomatic” (Table 1). Most isolates were isolated/recovered from urine samples. Symptomatic patients had a wide range of symptoms, from hearing loss to fetal loss. Our panel also includes multiple times of infection: among the asymptomatic neonates, two were infected during the second trimester and one during the third trimester; and among the symptomatic neonates, two were infected during the first trimester and one (deafness) during the third trimester. From these different clinical isolates, we were able to obtain free cell virus stocks with passages between four and eight.

### 3.2. CMV Genotyping of Clinical Strains

For each isolate, the full-length genome consensus sequence was used to perform genotyping and comparison. Genotypes were identified from the following genes encoding gB (*UL55*), gN (*UL73*), gO (*UL74*), gH (*UL75*), and gL (*UL115*) (Table 2 and Appendix A). Three of the 5 gB genotypes were represented in the symptomatic strain panels; gB1, gB2, and gB4, with gB1 being the most prevalent among all strains (67%). The gN4c genotype was the most represented in clinical strains, and two symptomatic strains showed a gN-5 genotype (2015-FR-02 and 2015-FR-04). GO-1c was the most common gO genotype among the strains (2/3 asymptomatic strains and 2/6 symptomatic strains). Thirty-three percent of the symptomatic strains had a gH2 genotype, while 100% of the asymptomatic strains had a gH1 genotype. We observed a predominance of the gL2 genotype (56%), while gL3 was absent from our panel.

Interestingly, the D309H mutation in the *UL122* gene, which is associated with TB40/E epithelial tropism [13], was present in all our isolates. We analyzed the influence of glycoprotein genes and virulence factor genes using principal component analysis (PCA) (see Figure 1 and Appendix A). We observed that symptomatic strains were segregated from asymptomatic strains on the genes studied; *UL115* and *UL100* partially explain this distribution. The asymptomatic strains, 2022-FR-02 and 2015-FR-06, which were acquired by primary infection, seem to be quite close to each other. The symptomatic strains, especially 2015-FR-03 (leading to hearing loss), 2015-FR-02, and 2015-FR-04 (leading to neurological defects), show more differences and are more distant in the figure. Asymptomatic strain 2022-FR-05 and symptomatic strain 2022-FR-03 (causing mild hearing loss), which were acquired in late secondary infection, are similar in terms of variation.

### 3.3. First Trimester Villi and Decidua Are Similarly Permissive to HCMV Laboratory Strains

We evaluated the replication of two laboratory strains, BAD-CMV and TB40/E, in first trimester placental explants. For both laboratory strains, infection on day 0 (plate switch day) and replication throughout the experiment were similar in villi and decidua, with a peak in viral load beginning on day 7–10 post-plate change. However, some behavioral differences were observed. BAD-CMV infects placental tissue better, with a geometric mean title of 7 log copies of CMV/million cells versus 5 log copies of CMV/million cells for TB40/E (Figure 2). Late first trimester villi (≥11 weeks) showed an increase of TB40/E viral load at day 10 post-plate switch (1 Log fold change between early and late tissue). However, as this difference is only statistically significant at this single time-point, the permissiveness of late placentas does not appear to be significantly different from early placentas.

### 3.4. Symptomatic and Asymptomatic HCMV Strains can Infect First Trimester Placentas

We aimed to determine the viral load kinetics in villi and decidua tissues infected with symptomatic and asymptomatic strains (Figure 3). We observed that the majority of symptomatic and asymptomatic strains are able to infect and replicate in villi, except for the strains 2015-FR-06 (asymptomatic), 2015-FR-04, and 2022-FR-04 (symptomatic), for which we could not detect any copies of HCMV up to 13 days after plate change. Compared to symptomatic strains, asymptomatic strains tend to replicate poorly in decidua tissue (range of 2–3 log/copies of CMV genome/million cells at day 13 after infection in decidua vs. a range of 3–5 log/copies of CMV genome/million cells at day 13 after infection in villi).

Interestingly, placental age seems to affect the infectivity of the symptomatic strain 2015-FR-01: Replication of this strain is no longer detected in ≥11 weeks villi and decidua.

## 4. Discussion

Congenital HCMV infection is a major problem and the mechanisms leading to symptomatic or asymptomatic infection remain unclear. We aimed to investigate patterns of HCMV placental infection at early stages of pregnancy by comparing infection by a panel of clinical asymptomatic and symptomatic strains to laboratory strains. Our study is the first to infect ex vivo placenta models with clinical isolates. This new approach allows for the comparison of clinical strain behavior with reference strains and to highlight differences between their kinetics both in villi and in decidua. In addition, these isolates have been well characterized by whole genome analysis at low passages.

Our work has shown that asymptomatic strains are able to infect both the villi and the decidua of the first trimester, but with a limited replication capacity in the decidua compared to symptomatic strains, which may explain their lower impact on infected newborns. We could not detect the replication of two symptomatic strains (2015-FR-04 and 2022-FR-04) and one asymptomatic strain (2015-FR-06) in first trimester tissues, while they were able to infect the monolayer fibroblasts, as shown by the presence of infectious foci. This could be due to a change of replicative capacity in fibroblasts following passages on ARPE-19 cells, leading to low secretion of virus in the culture supernatant, which does not allow sufficient infection of explants—or rather to a very low replication in placental tissues—below the detection threshold of our qPCR. We observed that one clinical strain infects and replicates only in placentas <11 weeks of amenorrhea, corresponding to the beginning of the second trimester of pregnancy. Placental development between the first and second trimesters involves several morphological changes, including the disappearance of trophoblastic plugs founded in the spiral arteries [14]; this may partially explain our results, and deserves further investigation.

Several studies have attempted to correlate glycoprotein genotypes with symptom severity in HCMV-infected neonates, with conflicting results [15,16,17]. Some variants for the UL130 and UL131 genes appear to have an impact on symptomatology in congenitally infected infants [18]. One strength of our work is the analysis of the whole genome of clinical isolates from the very first passages. All strains were sequenced before the fourth passage, after isolation, and were cultured in ARPE-19 cells to limit the emergence of mutations. Indeed, a previous study [19] showed that mutation emergence is less frequent in epithelial cell lines and occurs after passage 20. Moreover, the strains were used for placental assays between the fourth and eighth passage after isolation, thus limiting the risk of divergence during viral stock generation and subsequent experiments. Our strain panel included several glycoprotein genotypes, but none were associated with the onset of symptoms or the ability to replicate in the placenta. A PCA analysis of glycoprotein genes and virulence factors showed a segregation between asymptomatic and symptomatic strains, which was partly based on glycoproteins gM (*UL100*) and gL (*UL115*), while variations in *UL128*, *UL130*, or *UL131* were not significant. The gM coding sequence is highly conserved with only one epitope, and no genotypes could be defined for this gene [15]. Further analysis of this small protein, which forms a complex with gN, may be of interest.

The main limitation of our study is the size of our panel, with a low number of asymptomatic CMV, as well as a limited number of available placentas. However, very few laboratories routinely isolate congenital strains of CMV, and most samples are only suitable for molecular diagnosis. In addition, it is difficult to obtain virion stocks from low-passage isolates. The analysis of nine well-characterized isolates in this model, which requires selection of several placentas from seronegative women, was therefore challenging.

However, the overall results provide a first indication of the behavior of clinical strains during congenital HCMV infection. Our model can be further applied to a larger panel of strains and tissues, and further investigations on the genotypic analysis of clinical strains, with more clinical data on the mothers (co-infection, placental pathologies, etc.) could allow for a better understanding of the mechanisms of placental infection that do or do not lead to symptomatology.

## Figures and Tables

**Figure 1 pathogens-12-00985-f001:**
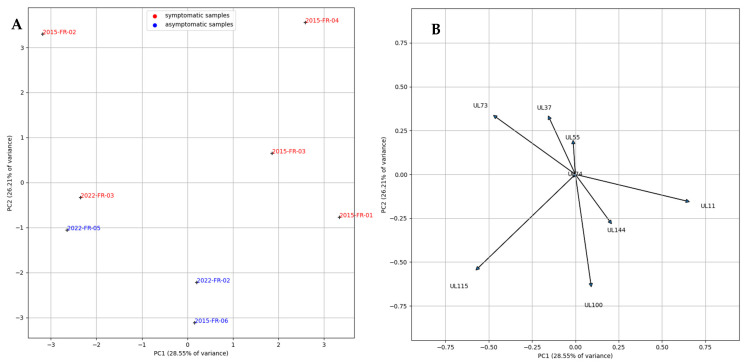
Principal component analysis (PCA) representing individual factor map for similarities based on variance of glycoprotein genes and virulence factors for all clinical strains. (**A**) =PCA plots. (**B**) =PCA variables. The number of mutations per gene and parallel coordinates chart are provided as Appendix A.

**Figure 2 pathogens-12-00985-f002:**
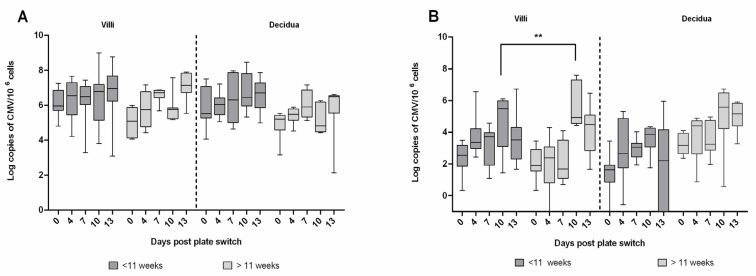
Kinetics of laboratory strains BAD-CMV and TB40/E in villi and decidua explants isolated from first-trimester placentas collected before (early) or after (late) 11 weeks of amenorrhea. (**A**) Kinetics of BAD-CMV in 5 early and 5 late placentas from different donors. (**B**) TB40/E kinetics in 7 early and 4 late placentas from different donors. The amount of HCMV in tissues is expressed as logarithms of CMV genome copies per million cells, and whiskers are extended to the extreme data points. **: *p* < 0.01 (two-way ANOVA test).

**Figure 3 pathogens-12-00985-f003:**
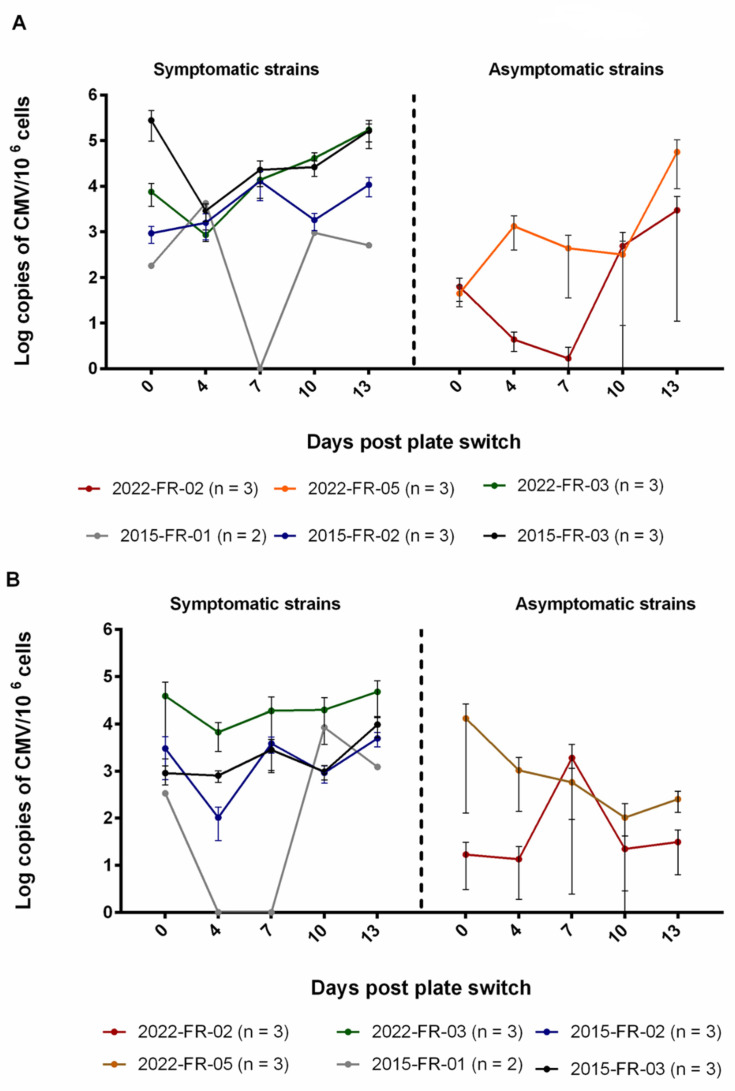
Kinetics of clinical strains in floating villi and decidua isolated from first trimester placenta. Data are presented as mean (3 replicates for each n placentas, from different donors) ± standard error of the mean (SEM). (**A**) Viral loads measured in villi. (**B**) Viral loads measured in the decidua. The amount of HCMV in tissues is expressed as Log of CMV genome copies per million cells. N = number of placentas tested.

**Table 1 pathogens-12-00985-t001:** Characteristics of clinical HCMV strains provided by NRC for herpesviruses, in terms of infection time during pregnancy, symptomatology, and number of passages after cell-free virus stock production. WA: weeks of amenorrhea. Nd: not determined. (S): Symptomatic strain, (A): asymptomatic strain * Secondary infection means CMV infection in a mother that was CMV-seropositive before pregnancy. Reinfections by a new CMV strain or reactivation of endogenous CMV cannot be distinguished.

Number	Sample Type	Infection Time	Symptoms	Number of Passages
2015-FR-01 (S)	Urine	>15 WA	Psychomotor retardation	4
2015-FR-02 (S)	Urine	Nd	Neurological defects	8
2015-FR-03 (S)	Urine	Nd	Hearing loss	5
2015-FR-04 (S)	Urine	Nd	Neurological defects	4
2022-FR-04 (S)	Biopsies	Primary infection 1st trimester	Fetus loss	5
2022-FR-03 (S)	Saliva	Secondary infection * >28 WA	Bilateral deafness	5
2015-FR-06 (A)	Urine	Primary infection 20–25 WA	None	6
2022-FR-02 (A)	Urine	Primary infection 12–17 WA	None	6
2022-FR-05 (A)	Urine	Secondary infection * 36–38 WA	None	5

**Table 2 pathogens-12-00985-t002:** Overview of glycoprotein genotypes of clinical strains determined after alignment against laboratory strain sequences. (S) = symptomatic, (A) = asymptomatic.

Strain Number	gB (*UL55*)	gN (*UL73*)	gO (*UL74*)	gH (*UL75*)	gL (*UL115*)
2015-FR-01 (S)	1	4c	5	2	2
2015-FR-02 (S)	1	5	1b	1	1
2015-FR-03 (S)	2	4c	5	2	2
2015-FR-04 (S)	1	5	1b	1	2
2022-FR-04 (S)	1	4c	1c	1	1
2022-FR-03 (S)	4	4b	1c	1	4
2015-FR-06 (A)	1	4c	1c	1	2
2022-FR-02 (A)	1	4c	1c	1	1
2022-FR-05 (A)	2	4a	1a	1	2

## Data Availability

Genomic analysis, clinical data and placental models results are avaible on demand to the corresponding authors at the National Reference Center.

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
