# Peer review of "Genetic and Functional Characterization of Congenital HCMV Clinical Strains in Ex Vivo First Trimester Placental Model"

_pathogens, 2023, doi:10.3390/pathogens12080985_

Round 1
Reviewer 1 Report
The authors use a reasonable approach to address the question why CMV may cause symptomatic congnital disease in some children whereas most remain asymptomatic. Transplacental CMV transmission is likely a critical process determining the clincal outcome of the unborn child and every factrot influencing this is of importance. Here, the authors addressed the question if various CMV strains account for the different infectivity but do exclude host factors that could be infvolved. This is totally fine for this report but should be clarified (see comment to figure 2). The manuscript is written coherent but data presentation should be improved.
190-201: Could you please comment on the genes of the pentameric complex?
lines 214-215: please specify in detail how "secondary infection" was distinguisehd from "seroconversion"
line 239: this sentence is not completed
Figure 2: please provide more information on what is depicted in the graphs. Dots represent different virus strains? How many different placentas have been used? How was statistical testing performed in detail?
Figure 3 and lines 250-253: if the authors want to make this strong statement the data should be presented according to the scientific question. Please perform adequate statistical testing and improve how to illustrate these results
Figure 4: the resolution is too low make any statement on fluorescence signal specificity and this figure needs to be improved
Figure 5: I simply cannot detect any green signal here and it is hard to discriminate autofluorescence of the tissue from specific antybody staining
lines 291-297: please be careful with these statements and use them only when data is well-presented in the results section. In the current version, I am not convinced.
English is comprehensible but should be improved.
Reviewer 2 Report
In this manuscript, the authors demonstrate a capacity to infect ex vivo placenta with a variety of HCMV strains. Given the restricted host nature of HCMV, being able to infect and analyse virus transmission within clinically relevant tissues is important. Having said that, the nature of the work, which involves both virus and host (placental) variation, does make robust conclusions difficult to draw. Nevertheless, the authors should be commended for attempting to move our cellular models of infection forwards.
Major comments:
The applicants use a variety of more clinical vs highly passaged strains in their work. The genetic analysis of the clinical strains is important, however although they passaged these strains sufficiently to induce cell-free virus production, they do not comment on which in vitro mutations were acquired during this process. Knowing these lesions is critical to interpret all of the functional work, and needs to be discussed throughout. I also wonder whether obvious acquired mutations should be reverted to their wildtype form for the PCA analysis?
The PCA is based on an analysis of a restricted number of glycoproteins, and a selection of ‘virulence’ genes – yet the rationale for selecting these genes is not given. The PCA does not appear to demonstrate massively strong clustering, making me wonder how sensitive it is to the selection of genes. Since the authors are investigating virus infection and spread, there is justification in basing it on entry glycoproteins. However, in that case it should include all entry glycoproteins (i.e. to also include gM, UL128, UL130, UL131A). The selection of ‘virulence factors’ is important, but the genes chosen are hard to rationalize – in a genome of ~170 ORFs, the vast majority of which are not required for in vitro replication, surely almost all genes are ‘virulence factors’? Even if they exclude genes of unknown function, there are far more factors with known functions in virulence than the subset used here? Finally, on the subject of the PCA, is it possible to determine which genes drive the segregation most strongly?
Minor points:
Line 206: the authors claim that the D309H mutation affects epithelial tropism, but as far as I can tell from the reference, it affects UL84 dependency of genome replication?
Line 212 – ‘mutations’ seem to be the wrong word here. Surely these are natural variations?
Line 234 – this section uses all laboratory strains, I think they have mistakenly called them clinical strains?
Fig 2 (and Fig 3) – I assume that each dot is a different placenta? Or is each dot a different virus? This needs clarifying. In addition, it would help if the same placenta/virus at each timepoint was connected by by lines, in the graph? Also, do the statistics take account of the fact that values are ‘linked/paired’ between the different timepoints?
Line 238 – the authors claim that late placenta are more permissive, but this is based on a single timepoint where this was true. For all other timepoints it was not true. On that basis, I don’t think that this conclusion is reasonable.
Fig 3 – there are a number of issues that need addressing on this graph. There are no statistics. It’s also very difficult to tell how many datasets are on the ‘zero’ line. Amongst the ‘symptomatic’ strains, no infection was detected for 3 strains, yet no values are shown in the ‘zero’ line – while for asymptomatic viruses, several are shown on the ‘zero’ line? I also wonder how much the averages are skewed by the viruses that showed no infection - what happens if all non-infectious viruses are removed?
Throughout, it’s not clear which samples were analysed in the same placental samples, and which were analysed in different placenta, and may therefore exhibit donor variation.
Some minor grammatical issues need correcting.
Reviewer 3 Report
In “Genetic and functional characterization of congenital HCMV clinical strains in ex vivo 1st trimester placental model,” Andouard and colleagues characterize the genetic differences between several clinical strains of cytomegalovirus that were recovered after congenital infections. Additionally, first trimester placental explants are infected with these strains and compared with parallel experiments that use common laboratory strains. This is the latest of a series of studies that have attempted to correlate viral strains and genomes with the outcomes of cCMV in children. The authors acknowledge some limitations of their study; notably, the number of samples analyzed is small and not representative of all CMV diversity. Even with this acknowledgement, there are serious deficiencies in the data analysis and presentation that undermine the authors’ conclusion that differences in replication kinetics and cell tropism were observed between the isolates.
The manuscript opens with a bioinformatic analysis of 9 clinical CMV isolates. Whole genomes were sequenced from low (<4) passage isolates. The authors should justify their decision to use a SARS-Cov-2 variant detection pipeline rather than tools specifically designed for HCMV genome analyses. While whole genome sequencing was done, the bioinformatics analysis focuses on select glycoprotein genes and a handful of virulence factors. Strains are genotyped based on the sequences of their glycoproteins but the original references for this classification system was not cited. Analyses that are referenced in the methods section (e.g. phylogenetic trees) are not shown. With the wealth of full-length CMV genomes available today, it is a missed opportunity to have not more compressively compared these 9 viruses with others. Glycoprotein and virulence factor sequences are compared in a principal component analysis, which attempts to correlate genome sequences with clinical presentation. The conclusion that samples have clustered in a way that reflects clinical symptoms is unconvincing.
Next, first trimester placenta and decidua are infected with HCMV ex vivo. For these experiments, fibroblasts are infected with HCMV at high multiplicity. Tissue samples are added 7 days later, and the tissue is transferred to a new plate after 7 more days. While this method has been previously published, it is somewhat unusual. Differences in the different replicative properties of the strains that occur in the fibroblasts will affect the later tissue explant infections, making it difficult to compare between the isolates. The abundance of viral genomes in tissue is quantified by qPCR after the tissue transferred to new plates. Additional detail is needed to interpret these experiments. In Figures 2 and 3, what does each data point represent? How many unique patient samples are represented in the data? Figure 3 appears to combine data from multiple strains isolated from symptomatic and asymptomatic children. This is inappropriate and the data from strains should be show individually so the reader can compare the properties of the different isolates.
Finally, differences in the cellular tropism of the different CMV isolates are reported after explants are infected and stained. The immunohistochemical analysis of infected placenta is not convincing. Representative images from select strains are shown, but the images are low resolution and appear to be taken at a low power. In Figure 4, strains should be explicitly identified. In Figure 5 it is not clear how the authors are detecting infected cells and specific cell types. Clarify how the tissue is being stained. A more substantive image analysis is necessary to illustrate that differences between the different strains exist.
The manuscript should be edited to improve the quality of the language as grammatical errors were fairly frequent.
Round 2
Reviewer 2 Report
I thank the authors for addressing the bulk of my questions. There are just two remaining issues - they did not address my very first question, which is to state what mutations were acquired during in vitro passage, since it is not possible to interpret downstream analyses without this. Secondly, Fig 3 is now much better, but should it not have error bars?
Author Response
I thank the authors for addressing the bulk of my questions. There are just two remaining issues - they did not address my very first question, which is to state what mutations were acquired during in vitro passage, since it is not possible to interpret downstream analyses without this. Secondly, Fig 3 is now much better, but should it not have error bars?
Dear Reviewer,
The aim of the sequence analysis was to adress the sequencing profile of the clinical isolates during the very first passages in culture. A study carried out on clinical and laboratory strains (Dargan et al,. 2010) showed the emergence, after 20 passages, of mutations in strains cultivated in epithelial cells, which is greater than the number of passages of our strains in ARPE-19. As our isolates are low-passaged and were cultivated in ARPE-19 before infection, with a global number of passages quite low (Table1), we stated that they were not significantly divergent. This was more clearly stated to the discussion.
We also have added the error bars in Fig3 as requested.
Reviewer 3 Report
The authors were responsive to reviewer feedback but concerns about data quality and interpretation in the revised manuscript remain.
· The bioinformatics analysis of viral genomes has been updated to include several more viral genes but is still lacking. Phylogenetic trees generated using the amino acid sequences of HCMV glycoproteins and a parallel coordinate chart were added as supplemental figures with little explanation. Supplemental Figure 1 is illegible when printed. This data would be better presented as phylograms rather than cladograms so the evolutionary distance between strains could be better inferred. The parallel coordinate chart needs to be explained to the reader.
· Reviewer 2 raised the concern that the viruses were all sequenced after 4-8 passages in ARPE-19 cells. While it may not be feasible at this time, it would be very informative if virus could be sequenced directly from the original clinical samples so that mutations that arose during passaging could be identified and distinguished from naturally occurring variations between different strains.
· In response to reviewer feedback, figure 3 was revised to show data from each strain that was tested. For each time point the data appears to represent 3 “placentas.” The authors should clarify whether these were explants from 3 different donors/time point or replicate explants from a single placenta. Error bars should also be added for each point to account for replicates. For consistency, Figure 2 could be updated so that data is presented the same was as it is in Figure 3.
· The immunostaining of villi and decidua is not convincing; signal may be tissue autofluorescence. Authors should clearly mark where decidua versus villi are shown. For villi-VIM signal should be restricted to the stromal core of villi and syncytiotrophoblasts should not be stained.
Minor points:
2022-FR-05 appears to be mislabeled as 2002-Fr-05 in several figures.
The manuscript should be edited to improve the quality of the language as grammatical errors were fairly frequent.
Round 3
Reviewer 3 Report
In the revised manuscript, the authors have provided additional details on bioinformatics analyses and improved some related figures. Concerns about the quality of epifluorescent micrographs remain. The authors have acknowledged that tissue autofluorescence confounds data interpretation and referenced another recently published manuscript where the same general approach yielded cleaner results. It is unfortunate that additional slides are not available that would enable the authors to repeat their experiment, but this does not justify the publication of data of suspect quality.
Manuscript would benefit from proofing to improve grammar.
Author Response
We thank the reviewer for his review that improves the paper and for his comment. We agree that our slides are not of good enough quality to be published. We have therefore modified the manuscript to take account of your comments, and we preferred to delete the sections concerning immunofluorescence. We have also proofread the article and harmonized the English grammar.
We hope that these modifications will fulfill your requirements and that the paper is now suitable for publication.